# Effective Diversity in Population Based Reinforcement Learning

**Jack Parker-Holder**\*
University of Oxford
jackph@robots.ox.ac.uk

**Aldo Pacchiano**\*
UC Berkeley
pacchiano@berkeley.edu

**Krzysztof Choromanski**
Google Brain Robotics
kchoro@google.com

**Stephen Roberts**
University of Oxford
sjrob@robots.ox.ac.uk

## Abstract

Exploration is a key problem in reinforcement learning, since agents can only learn from data they acquire in the environment. With that in mind, maintaining a population of agents is an attractive method, as it allows data be collected with a diverse set of behaviors. This behavioral diversity is often boosted via multi-objective loss functions. However, those approaches typically leverage mean field updates based on pairwise distances, which makes them susceptible to cycling behaviors and increased redundancy. In addition, explicitly boosting diversity often has a detrimental impact on optimizing already fruitful behaviors for rewards. As such, the reward-diversity trade off typically relies on heuristics. Finally, such methods require behavioral representations, often handcrafted and domain specific. In this paper, we introduce an approach to optimize all members of a population simultaneously. Rather than using pairwise distance, we measure the volume of the entire population in a behavioral manifold, defined by task-agnostic behavioral embeddings. In addition, our algorithm *Diversity via Determinants* (DvD), adapts the degree of diversity during training using online learning techniques. We introduce both evolutionary and gradient-based instantiations of DvD and show they effectively improve exploration without reducing performance when better exploration is not required.

## 1 Introduction

Reinforcement Learning (RL) considers the problem of an agent taking actions in an environment to maximize total (discounted/expected) reward [59]. An agent can typically only learn from experience acquired in the environment, making exploration crucial to learn high performing behaviors.

Training a *population* of agents is a promising approach to gathering a diverse range of experiences, often with the same (wall-clock) training time [44, 32]. Population-based methods are particularly prominent in the Neuroevolution community [56], but have recently been of increased interest in RL [23, 33, 25, 23, 24, 42]. One particularly exciting class of neuroevolution methods is Quality Diversity (QD, [43]) where algorithms explicitly seek high performing, yet diverse behaviors [10]. However, these methods often have a few key shortcomings.

Typically, each agent is optimized with a mean field assumption, only considering its *individual* contribution to the population's joint reward-diversity objective. Consequently, cycles may arise, whereby different members of the population constantly switch between behaviors. This may prevent

---

any single agent exploiting a promising behavior. This phenomenon motivates the MAP-Elites algorithm [37, 10], whereby only one solution may lie in each quadrant of a pre-defined space.

This pre-defined space is a common issue with QD [26]. Behavioral characterizations (BCs) often have to be provided to the agent, for example, in locomotion it is common to use the final $(x, y)$ coordinates. As such, automating the discovery of BCs is an active research area [16]. In RL, gradients are usually taken with respect to the actions taken by an agent for a set of sampled states. This provides a natural way to embed a policy behavior [23, 21, 11]. Incidentally, the geometry induced by such embeddings has also been used in popular trust region algorithms [53, 52].

In this paper we formalize this approach and define *behavioral embeddings* as the actions taken by policies. We measure the diversity of the entire population as the volume of the inter-agent kernel (or similarity) matrix, which we show has theoretical benefits compared to pairwise distances. In our approach agents are still optimizing for their *local* rewards and this signal is a part of their hybrid objective that also takes into account the *global* goal of the population - its diversity.

However, we note that it remains a challenge to set the diversity-reward trade off. If misspecified, we may see fruitful behaviors disregarded. Existing methods either rely on a (potentially brittle) hyperparameter, or introduce an annealing-based heuristic [9]. To ensure diversity is promoted *effectively*, our approach is to adapt the diversity-reward objective using Thompson sampling [46, 47]. This provides us with a principled means to trade-off reward vs. diversity through the lens of multi-armed bandits [54]. Combining these insights, we introduce Diversity via Determinants (DvD).

DvD is a general approach, and we introduce two implementations: one building upon Evolution Strategies, DvD-ES, which is able to discover diverse solutions in multi-modal environments, and an off-policy RL algorithm, DvD-TD3, which leads to greater performance on the challenging Humanoid task. Crucially, we show DvD still performs well when diversity is not required.

This paper is organized as follows. **(1)** We begin by introducing main concepts in Sec. 2 and providing strong motivation for our DvD algorithm in Sec. 3. **(2)** We then present our algorithm in Sec. 4. **(3)** In Sec. 5 we discuss related work in more detail. **(4)** In Sec. 6 we provide empirical evidence of the effectiveness of DvD. **(5)** We conclude and explain broader impact in Sec. 7 and provide additional technical details in the Appendix (including ablation studies and proofs).

## 2 Preliminaries

A Markov Decision Process (MDP) is a tuple $(\mathcal{S}, \mathcal{A}, \mathrm{P}, \mathrm{R})$. Here $\mathcal{S}$ and $\mathcal{A}$ stand for the sets of states and actions respectively, such that for $s_t, s_{t+1} \in \mathcal{S}$ and $a_t \in \mathcal{A}$: $\mathrm{P}(s_{t+1}|s_t, a_t)$ is the probability that the system/agent transitions from $s_t$ to $s_{t+1}$ given action $a_t$ and $r_t = r(s_t, a_t, s_{t+1})$ is a reward obtained by an agent transitioning from $s_t$ to $s_{t+1}$ via $a_t$.

A policy $\pi_\theta : \mathcal{S} \to \mathcal{A}$ is a (possibly randomized) mapping (parameterized by $\theta \in \mathbb{R}^d$) from $\mathcal{S}$ to $\mathcal{A}$. Policies $\pi_\theta$ are typically represented by neural networks, encoded by parameter vectors $\theta \in \mathbb{R}^d$ (their flattened representations). Model free RL methods are either on-policy, focusing on gradients of the policy from samples [52, 53], or off-policy, focusing on learning a value function [35, 15, 19].

In on-policy RL, the goal is to optimize parameters $\theta$ of $\pi_\theta$ such that an agent deploying policy $\pi_\theta$ in the environment given by a fixed MDP maximizes $R(\tau) = \sum_{t=1}^{T} r_t$, the total (expected/discounted) reward over a rollout time-step horizon $T$ (assumed to be finite). The typical objective is as follows:

$$J(\pi_\theta) = \mathbb{E}_{\tau \sim \pi_\theta}[R(\tau)] \qquad (1)$$

where $P(\tau|\theta) = \rho(s_0) \prod_{t=1}^{T} P(s_{t+1}|s_t, a_t)\pi_\theta(a_t|s_t)$, for initial state probability $\rho(s_0)$ and transition dynamics $P(s_{t+1}|s_t, a_t)$, which is often deterministic. Policy gradient (PG) methods [53, 52], consider objectives of the following form:

$$\nabla_\theta J(\pi_\theta) = \mathbb{E}_{\tau \sim \pi_\theta}\left[\sum_{t=0}^{T} \nabla_\theta \log \pi_\theta(a_t|s_t) R(\tau)\right], \qquad (2)$$

which can be approximated with samples (in the action space) by sampling $a_t \sim \pi_\theta(a_t|s_t)$.

An alternative formulation is Evolution Strategies (ES, [49]), which has recently shown to be effective [58, 7, 34]. ES methods optimize Equation 1 by considering $J(\pi_\theta)$ to be a blackbox function

$F : \mathbb{R}^d \to \mathbb{R}$ taking as input parameters $\theta \in \mathbb{R}^d$ of a policy $\pi_\theta$ and outputting $R(\tau)$. One benefit of this approach is potentially achieving deep exploration [41, 14].

A key benefit of ES methods is they naturally allow us to maintain a population of solutions, which has been used to boost diversity. Novelty search methods [9, 31] go further and explicitly augment the loss function with an additional term, as follows:

$$J(\pi_\theta) = R(\tau_\theta) + \lambda d(\tau_\theta) \tag{3}$$

where $\lambda > 0$ is the reward-diversity trade-off. We assume the policy and environment are deterministic, so $\tau_\theta$ is the only possible trajectory, and $d(\tau_{\theta^i}) = \frac{1}{M} \sum_{j \in M, j \neq i} ||BC(\pi_\theta^i) - BC(\pi_\theta^j)||_2$ for some $l$-dimensional behavioral mapping $BC : \tau \to \mathbb{R}^l$.

This formulation has been shown to boost exploration, and as such, there have been a variety of attempts to incorporate novelty-based metrics into RL algorithms [23, 11, 21]. Ideally, we would guide optimization in a way which would evenly distribute policies in areas of the embedding space, which correspond to high rewards. However, the policy embeddings (BCs) used are often based on heuristics which may not generalize to new environments [26]. In addition, the single sequential updates may lead high performing policies away from an improved reward (as is the case in the Humanoid experiment in [9]). Finally, cycles may become present, whereby the population moves from one area of the feature space to a new area and back again [37].

Below we address these issues, and introduce a new objective which updates all agents simultaneously.

## 3 Diversity via Determinants

Here we introduce our task agnostic embeddings and formalize our approach to optimize for population-wide diversity.

### 3.1 Task Agnostic Behavioral Embeddings

Many popular policy gradient algorithms [52, 53] consider a setting whereby a new policy $\pi_{\theta_{t+1}}$ is optimized with a constraint on the size of the update. This requirement can be considered as a constraint on the behavior of the new policy [40]. Despite the remarkable success of these algorithms, there has been little consideration for action-based behavior embeddings for novelty search methods.

Inspired by this approach, we choose to represent policies as follows:

**Definition 3.1.** *Let $\theta^i$ be a vector of neural network parameters encoding a policy $\pi_{\theta^i}$ and let $\mathcal{S}$ be a finite set of states. The **Behavioral Embedding** of $\theta^i$ is defined as: $\phi(\theta^i) = \{\pi_{\theta^i}(.|s)\}_{s \in \mathcal{S}}$.*

This approach allows us to represent the behavior of policies in vectorized form, as $\phi : \theta \to \mathbb{R}^l$, where $l = |a| \times N$, where $|a|$ is the dimensionality of each action $a \in \mathcal{A}$ and $N$ is the total number of states. When $N$ is the number of states in a finite MDP, the policies are determimistic and the embedding is the as in Definition 3.1, we have:

$$\phi(\theta^i) = \phi(\theta^j) \iff \pi_{\theta^i} = \pi_{\theta^j}, \tag{4}$$

where $\theta^i, \theta^j$ are vectorized parameters. In other words, the policies are the same since they always take the same action in every state of the finite MDP. Note that this does not imply that $\theta^i = \theta^j$.

### 3.2 Joint Population Update

Equipped with this notion of a behavioral embedding, we can now consider a means to compare two policies. Consider a smooth kernel $k$, such that $k(x_1, x_2) \leq 1$, for $x_1, x_2 \in \mathbb{R}^d$. A popular choice of kernel is the squared exponential (SE), defined as follows:

$$k_{\text{SE}}(x_1, x_2) = \exp\left(-\frac{||x_1 - x_2||^2}{2l^2}\right), \tag{5}$$

for some length-scale $l > 0$. Now moving back to policy embeddings, we extend our previous analysis to the kernel or similarity between two embeddings, as follows:

$$k(\phi(\theta^i), \phi(\theta^j)) = 1 \iff \pi_{\theta^i} = \pi_{\theta^j} \tag{6}$$

We consider two policies to be behaviorally orthogonal if $k(\phi(\theta^i), \phi(\theta^j)) = 0$. With a flexible way of measuring inter-policy similarity at hand, we can define the *population-wide* diversity as follows:

**Definition 3.2.** *(Population Diversity) Consider a finite set of $M$ policies, parameterized by $\Theta = \{\theta^1, ..., \theta^M\}$, with $\theta^i \in \mathbb{R}^d$. We denote $\mathrm{Div}(\Theta) \stackrel{\mathrm{def}}{=} \det(K(\phi(\theta_t^i), \phi(\theta_t^j))_{i,j=1}^M) = \det(\mathbf{K})$, where $K : \mathbb{R}^l \times \mathbb{R}^l \to \mathbb{R}$ is a given kernel function. Matrix $\mathbf{K}$ is positive semidefinite since all principal minors of $\det(\mathbf{K})$ are nonnegative.*

This formulation is heavily inspired by Determinantal Point Processes (DPPs, [28]), a mechanism which produces diverse subsets by sampling proportionally to the determinant of the kernel matrix of points within the subset. From a geometric perspective, the determinant of the kernel matrix represents the volume of a parallelepiped spanned by feature maps corresponding to the kernel choice. We seek to maximize this volume, effectively "filling" the feature (or behavior) space.

Now consider the DvD loss function, as follows:

$$J(\Theta_t) = \underbrace{\sum_{i=1}^{M} \mathbb{E}_{\tau \sim \pi_{\theta^i}}[R(\tau)]}_{\text{individual rewards}} + \underbrace{\lambda_t \mathrm{Div}(\Theta_t)}_{\text{population diversity}} \tag{7}$$

where $\lambda_t \in (0, 1)$ is the trade-off between reward and diversity. This fundamentally differs from Equation 3, since we directly optimize for $\Theta_t$ rather than separately considering $\{\theta^i\}_{i=1}^M$.

**Theorem 3.3.** *Let $M$ be a finite, tabular MDP with $\tilde{M} \geq M$ distinct optimal policies $\{\pi_i\}_{i=1}^{\tilde{M}}$ all achieving a cumulative reward of $\mathcal{R}$ and such that the reward value $\mathcal{R}(\pi)$ of any suboptimal policy $\pi$ satisfies $\mathcal{R}(\pi) + \Delta < \mathcal{R}$ for some $\Delta > 0$. There exists $\lambda_t > 0$, such that the objective in Equation 7 can only be maximized if the population contains $M$ distinct optimal solutions.*

The proof of Theorem 3.3 is in the Appendix (Sec. 9), where we also show that for the case of the squared exponential kernel, the first order approximation to the determinant is in fact related to the mean pairwise L2-distance. However, for greater population sizes, this first order approximation is zero, implying the determinant comes from higher order terms.

We have now shown theoretically that our formulation for diversity can recover diverse high performing solutions. This result shows the importance of utilizing the determinant to measure diversity rather than the pairwise distance.

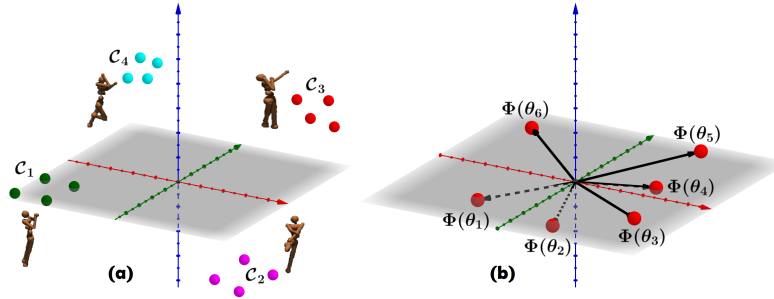

Figure 1: Determinant vs. pairwise distance. (a): populations of agents split into four clusters with agents within cluster discovering similar policies. (b): embedded policies $\phi(\theta_1), ..., \phi(\theta_6)$ lie in a grey hyperplane. In (a) resources within a cluster are wasted since agents discover very similar policies. In (b) all six embeddings can be described as linear combinations of embeddings of fewer canonical policies. In both settings the mean pairwise distance will be high but diversity as measured by determinants is low.

Using the determinant to quantify diversity prevents the undesirable clustering phenomenon, where a population evolves to a collection of conjugate classes. To illustrate this point, consider a simple scenario, where all $M$ agents are partitioned into $k$ clusters of size $\frac{M}{k}$ each for $k = o(M)$. By increasing the distance between the clusters one can easily make the novelty measured as an average distance between agents' embedded policies as large as desired, but that is not true for the corresponding determinants which will be zero if the similarity matrix is low rank. Furthermore, even if all pairwise distances are large, the determinants can be still close to zero if spaces where agents' high-dimensional policy embeddings live can be accurately approximated by much lower dimensional ones. Standard methods are too crude to measure novelty in such a way (see: Fig. 1).

Next we provide a simple concrete example to demonstrate this phenomenon.

### 3.3 An Illustrative Example: Tabular MDP

Consider the simple MDP in Fig. 2. There are four states $\mathcal{S} = \{s_0, s_1, s_2, s_3\}$, each has three actions, $\mathcal{A} = \{-1, 0, 1\}$ corresponding to left, down and right respectively. In addition, there are five terminal states, three of which achieve the maximum reward (+1). Let $\phi^* = \{\phi(\theta^i)\}_{i=1}^5 = \{[-1, -1], [-1, 1], [0, 0], [1, -1], [1, 1]\}$, be the set of optimal policies. If we have a population of five agents, each achieving a positive reward, the determinant of the $5 \times 5$ kernel matrix is only $> 0$ if the population of agents is exactly

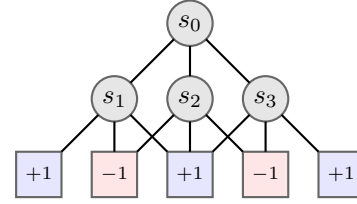

Figure 2: A simple MDP.

$\phi^*$. This may seem trivial, but note the same is not true for the pairwise distance. If we let $d(\Theta) = \frac{1}{n} \sum_{i=1}^n \sum_{j>i} ||\phi(\theta^i) - \phi(\theta^j)||_2$, then $\phi^*$ does not maximize $d$. One such example which achieves a higher value would be $\phi' = \{[-1, -1], [-1, 1], [1, -1], [1, 1], [1, 1]\}$. See the following link for a colab demonstration of this example: https://bit.ly/2XAlirX.

## 4 DvD Algorithm

### 4.1 Approximate Embeddings

In most practical settings the state space is intractably or infinitely large. Therefore, we must sample the states $\{s_i\}_{i=1}^n$, where $n < N$, and compute the embedding as an expectation as follows:

$$\widehat{\phi}(\theta_i) = \mathbb{E}_{s \sim \mathcal{S}}[\{\pi_{\theta^i}(.|s)\}] \tag{8}$$

In our experiments we choose to randomly sample the states $s$, which corresponds to frequency weights. Other possibilities include selecting diverse ensembles of states via DPP-driven sampling or using probabilistic models to select less frequently visited states. We explore each of these options in the experiments where we show the representative power of this action-based embedding is not overly sensitive to these design choices (see: Fig. 8). However, it remains a potential degree of freedom for future improvement in our method, potentially allowing a far smaller number of states to be used.

### 4.2 Adaptive Exploration

Optimizing Equation 7 relies on a user-specified degree of priority for each of the two objectives ($\lambda_t$). We formalize this problem through the lens of multi-armed bandits, and adaptively select $\lambda_t$ such that we encourage favoring the reward or diversity at different stages of optimization.

Specifically, we use Thompson Sampling [60, 55, 1, 46, 47]. Let $\mathcal{K} = \{1, \cdots, K\}$ denote a set of arms available to the decision maker (learner) who is interested in maximizing its expected cumulative reward. The optimal strategy for the learner is to pull the arm with the largest mean reward. At the beginning of each round the learner produces a sample mean from its mean reward model for each arm, and pulls the arm from which it obtained the largest sample. After observing the selected arm's reward, it updates its mean reward model.

Let $\pi_t^i$ be the learner's reward model for arm $i$ at time $t$. When $t = 0$ the learner initializes each of its mean reward models to prior distributions $\{\pi_0^i\}_{i=1}^K$. At any other time $t > 0$, the learner starts by sampling mean reward candidates $\mu_i \sim \pi_{t-1}^i$ and pulling the arm: $i_t = \arg \max_{i \in \mathcal{K}} \mu_i$. After observing a true reward sample $r_t$ from arm $i_t$, the learner updates its posterior distribution for arm $i_t$. All the posterior distributions over arms $i \neq i_t$ remain unchanged.

In this paper we make use of a Bernoulli model for the reward signal corresponding to the two arms ($\lambda = 0, \lambda = 0.5$). At any time $t$, the chosen arm's sample reward is the indicator variable $r_t = \mathbf{1}(R_{t+1} > R_t)$ where $R_t$ denotes the reward observed at $\theta_t$ and $R_{t+1}$ that at $\theta_{t+1}$. We make a simplifying stationarity assumption and disregard the changing nature of the arms' means in the course of optimization. We use beta distributions to model both the priors and the posteriors of the arms' means. For a more detailed description of the specifics of our methodology please see the Appendix (Sec. 10.1).

We believe this adaptive mechanism could also be used for count-based exploration methods or intrinsic rewards [51], and note very recent work using a similar approach to vary exploration in off-policy methods [50, 3] and model-based RL [4]. Combining these insights, we obtain the DvD algorithm. Next we describe two practical implementations of DvD.

### 4.3 DvD-ES Algorithm

At each timestep, the set of policies $\Theta_t = \{\theta_t^i\}_{i=1}^M$ are simultaneously perturbed, with rewards computed locally and diversity computed globally. These two objectives are combined to produce a blackbox function with respect to $\Theta_t$. At every iteration we sample $Mk$ Gaussian perturbation vectors $\{\mathbf{g}_i^m\}_{i=1,\ldots,k}^{m=1,\ldots,M}$. We use two partitionings of this $Mk$-element subset that illustrate our dual objective - high local rewards and large global diversity. The first partitioning assigns to $m^{\text{th}}$ worker a set $\{\mathbf{g}_1^m, \ldots, \mathbf{g}_k^m\}$. These are the perturbations used by the worker to compute its local rewards. The second partitioning splits $\{\mathbf{g}_i^m\}_{i=1,\ldots,k}^{m=1,\ldots,M}$ into subsets: $\mathcal{D}_i = \{\mathbf{g}_i^1, \ldots, \mathbf{g}_i^M\}$. Instead of measuring the contribution of an individual $\mathbf{g}_i^m$ to the diversity, we measure the contribution of the entire $\mathcal{D}_i$. This motivates the following definition of diversity:

$$\text{Div}_t(i) = \text{Div}_t(\theta_t^1 + \mathbf{g}_i^1, \ldots, \theta_t^M + \mathbf{g}_i^M). \tag{9}$$

Thus, the DvD-ES gradient update is the following:

$$\theta_{t+1}^m = \theta_t^m + \frac{\eta}{k\sigma} \sum_{i=1}^k [(1 - \lambda_t)R_i^m + \lambda_t \text{Div}_t(i)]\mathbf{g}_i^m. \tag{10}$$

where $\sigma > 0$ is the smoothing parameter [38, 49], $k$ is the number of ES-sensings, $\eta$ is the learning rate, and the embeddings are computed by sampling states from the most recent iteration.

### 4.4 DvD-TD3 Algorithm

It is also possible to compute analytic gradients for the diversity term in Equation 7. This means we can update policies with respect to the joint diversity using automatic differentiation.

**Lemma 4.1.** *The gradient of* $\log(\det(\mathbf{K}))$ *with respect to* $\Theta = \theta^1, \cdots, \theta^M$ *equals:* $\nabla_\theta \log(\det(\mathbf{K})) = -(\nabla_\theta \Phi(\theta))(\nabla_\Phi \mathbf{K})\mathbf{K}^{-1}$, *where* $\phi(\theta) = \phi(\theta^1) \cdots \phi(\theta^M)$.

The proof of this lemma is in the Appendix, Sec. 9.2. Inspired by [23], we introduce DvD-TD3, using multiple policies to collect data for a shared replay buffer. This is done by dividing the total data collection by the number of policies. When optimizing the policies, we use an augmented loss function, as in Equation 7, and make use of the samples in the existing batch for the embedding.

## 5 Related Work

Neuroevolution methods [56], seek to maximize the reward of a policy through approaches strongly motivated by natural biological processes. They typically work by perturbing a policy, and either computing a gradient (as in Evolution Strategies) or selecting the top performing perturbations (as in Genetic Algorithms). The simplicity and scalability of these methods have led to their increased popularity in solving RL tasks [49, 7, 6, 9, 58].

Neuroevolution methods often make use of behavioral representations [9, 17]. In [9] it is proposed to use a population of agents, each of which would seek to jointly maximize the reward and difference/*novelty* in comparison to other policies, quantified as the mean pairwise distance. Another approach, Evolvabilty ES [17] seeks to learn policies which can quickly adapt to a new task, by maximizing the variance or entropy of behaviors generated by a small perturbation. The MAP-Elites [37] algorithm is conceptually similar to ours, the authors seek to find quality solutions in differing dimensions. However, these dimensions need to be pre-specified, whereas our method can be considered a learned version of this approach. To the best of our knowledge, only one recent neuroevolutionary approach [22] uses the actions of a policy to represent behaviors, albeit in a genetic context over discrete actions.

In the RL community there has recently been interest in unsupervised learning of diverse behaviors [12, 20]. These methods are similar in principle to novelty search without a reward signal, but instead focus on diversity in behaviors defined by the states they visit. Another class of algorithms making use of behavioral representations [18] focuses on meta learning in the behavioral space, however they require pre-training on similar tasks in order to learn a new one. A meta learning approach from [48] proposes using a latent generative representation of model parameters, which could be thought of as a behavioral embedding of the policy. Finally, [61] propose a functional approach for learning diversified parameters. As an iterative method, it is still subject to the undesirable cycling phenomenon.

# 6 Experiments

Here evaluate DvD-ES and DvD-TD3 in a variety of challenging settings. We focus first on the ES setting, since ES is cheap to run on CPUs [49] which allows us to run a series of ablation studies.

## 6.1 Finding Diverse Solutions with DvD-ES

We compare DvD-ES against vanilla ES, as well as NSR-ES from [9], both of which updates each population member sequentially. For both DvD and NSR-ES, we use the same embedding, with 20 randomly selected states. We use this for all **all** DvD-ES experiments. We parameterize our policies with two hidden layer neural networks, with tanh activations (more details are in the Appendix, Section 8.2). All x-axes are presented in terms of iterations and comparisons are made fair by dividing the number of iterations for two sequential algorithms (ES and NSR-ES) by the population size. All experiments made use of the ray [36] library for parallel computing, with experiments run on a 32-core machine. To run these experiments see the following repo: https://github.com/jparkerholder/DvD_ES.

**Exploration** We begin with a simple environment, whereby a two dimensional point agent is given a reward equal to the negative distance away from a goal. The agent is separated from its goal by the wall (see: Fig. 3a)). A reward of -800 corresponds to moving directly towards the goal and hitting at the wall. If the agent goes around the wall it initially does worse, until it learns to arc towars the goal, which produces a reward greater than -600. We ran ten seeds,

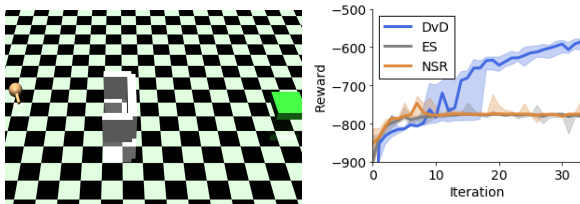

Figure 3: **Left:** the Point-v0 environment. **Right:** median best performing agent across ten seeds. IQR shaded.

with a population of size $M = 5$. As we see, both vanilla ES and NSR-ES fail to get past the wall (a reward of -800), yet DvD is able to solve the environment for **all 10 seeds**. See the Appendix (Table 4) for additional hyperparameters details.

**Multi-Modal Environments** A key attribute of a QD algorithm is the ability to learn a diverse set of high performing solutions. This is often demonstrated qualitatively, through videos of learned gaits, and thus hard to scientifically prove. To test this we create environments where an agent can be rewarded for multiple different behaviors, as is typically done in multi-objective RL [62]. Concretely, our environments are based on the Cheetah and Ant, where we assign rewards for *both* Forward and Backward tasks, which commonly used in meta-RL [13, 45]. We can then evaluate the population of agents on both individual tasks, to quantitatively evaluate the diversity of the solutions.

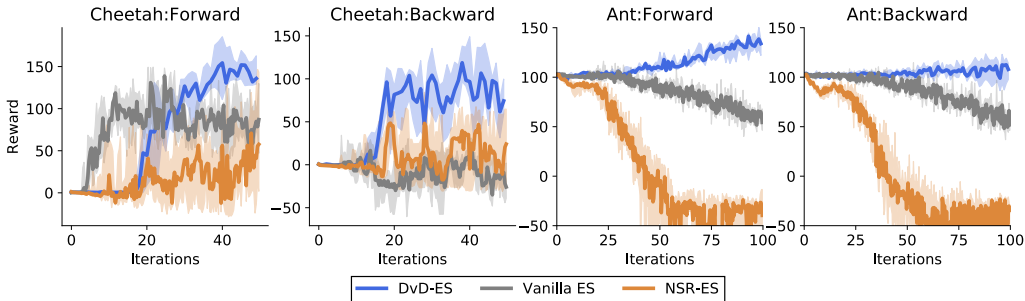

Figure 4: The median best performing agent across ten seeds for multi-modal tasks and two environments: Cheetah and Ant. The plots show median curves with IQR shaded.

In both settings we used a population size of $M = 3$, which is sufficient to learn both tasks. In Fig. 4 we see that DvD is able to learn both modes in both environments. For the Cheetah, the Backward task appears simpler to learn, and with no diversity term vanilla ES learns this task quickly, but subsequently performs poorly in the Forward task (with inverse reward function). For Ant, the noise from two separate tasks makes it impossible for vanilla ES to learn at all.

**Single Mode Environments** Now we consider the problem of optimizing tasks which have only one optimal solution or at least, with a smaller distance between optimal solutions (in the behavioral space), and an informative reward function. In this case, overly promoting diversity may lead to worse performance on the task at hand, as seen in NSR-ES ([9], Fig. 1.(c)). We test this using four widely studied continuous control tasks from OpenAI Gym. In all cases we use a population size of $M = 5$, we provide additional experimental details in the Appendix (see Section 8.2).

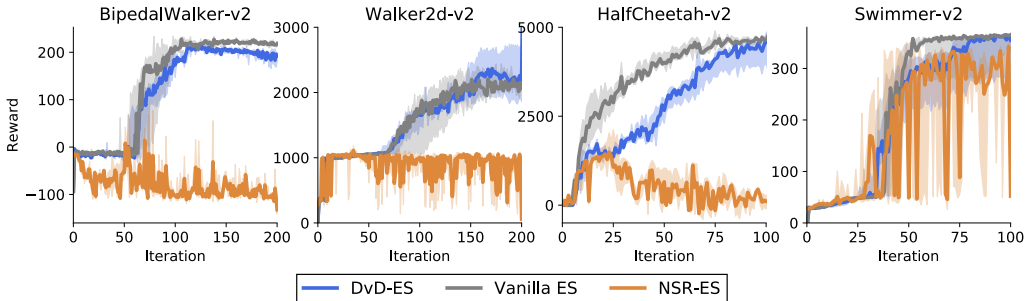

Figure 5: The median best performing agent across five seeds for four different environments.

As we see in Fig. 5, in all four environments NSR-ES fails to train (which is consistent with the vanilla Humanoid experiment from [9]) in contrast to DvD that shows only a minimal drop in performance vs. the vanilla ES method which is solely focusing on the reward function. This promising result implies that we gain the benefit of diversity without compromising on tasks, where it is not required. We note that DvD outperforms ES in Walker2d, which is known to have a deceptive local optimum at 1000 induced by the survival bonus [34].

These experiments enable us also to demonstrate the cyclic behaviour that standard novelty search approaches suffer from (see: Section 1). NSR-ES often initially performs well, before subsequently abandoning successful behaviors in the pursuit of novelty. Next we explore key design choices.

**Choice of kernel** In our experiments, we used the Squared Exponential (or RBF) kernel. This decision was made due to its widespread use in the machine learning community and many desirable properties. However, in order to assess the quality of our method it is important to consider the sensitivity to the choice of kernel. In Table 1 we show the median rewards achieved from 5 seeds for a variety of kernels. As we see, in almost all cases the performance is strong, and similar to the Squared Exponential kernel used in the main experiments.

Table 1: This table shows the median maximum performing agent across 5 seeds. All algorithms shown are identical aside from the choice of DPP kernel. Training is for $\{50, 100, 200\}$ iterations for $\{$point, SwimmerWalker2d$\}$ respectively.

|  | Point | Swimmer | Walker2d |
|---|---|---|---|
| Squared Exponential | -547.03 | 354.86 | 1925.86 |
| Exponential | -561.13 | 362.83 | 1929.81 |
| Linear | -551.48 | 354.37 | 1944.95 |
| Rational Quadratic | -548.55 | 246.68 | 2113.02 |
| Matern $\frac{3}{2}$ | -578.05 | 349.52 | 1981.66 |
| Matern $\frac{5}{2}$ | -557.69 | 357.88 | 1866.56 |

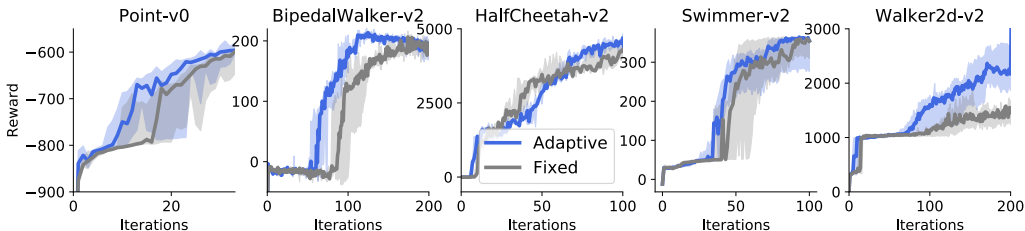

Figure 6: The median best performing agent across five seeds for five different environments.

**Do we need to adapt?** In Fig. 6 we explored the effectiveness of the adaptive mechanism, by running five experiments with the DvD algorithm with fixed $\lambda = 0.5$. While notably we still see strong performance from the joint diversity score (vs. NSR-ES in Fig 5), it is clear the adaptive mechanism boosts performance in all cases.

## 6.2 Teaching a Humanoid to Run with DvD-TD3

We now evaluate DvD-TD3 on the challenging Humanoid environment from the Open AI Gym [5]. The Humanoid-v2 task requires a significant degree of exploration, as there is a well-known local optimum at 5000, since the agent is provided with a survival bonus of 5 per timestep [34].

We train DvD-TD3 with $M = 5$ agents, where each agent has its own neural network policy, but a shared Q-function. We benchmark against both a single agent ($M = 1$), which is vanilla TD3, and then what we call ensemble TD3 (E-TD3) where $M = 5$ but there is no diversity term. We initialize all methods with $25,000$ random timesteps, where we set $\lambda_t = 0$ for DvD-TD3. We train each for a total of one million timesteps, and repeat for 7 seeds. The results are shown in Fig. 7.

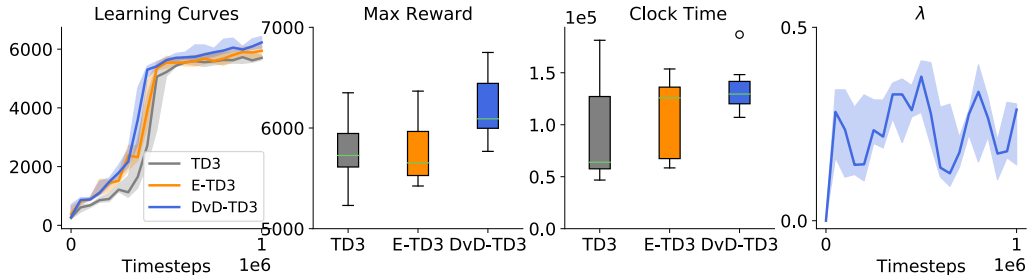

Figure 7: From left to right: a) Median curves from 7 seeds, IQR shaded. b) The distribution of maximum rewards achieved for each seed. c) The distribution of wall clock time to train for $10^6$ timesteps. d) The evolution of $\lambda_t$ during training.

As we see on the left two plots, DvD-TD3 achieves better sample efficiency, as well as stronger final performance. For comparison, the median best agents for each algorithm were: DvD-TD3: 6091, E-TD3: 5654 and TD3: 5727. This provides t-statistics of 2.35 and 2.29 for DvD-TD3 vs. E-TD3 and TD3, using Welch's unequal variances t-test. Both of these are statistically significant ($p < 0.05$). As far as we are aware, DvD-TD3 is the first algorithm to get over 6000 on Humanoid-v2 in one million timesteps. This means the agent has achieved forward locomotion, rather than simply standing still (at 5000). Previous results for Soft Actor Critic (SAC) have only reached 6000 at three million timesteps [19, 8]. We believe this provides strong support for using population-based approaches to explore with different behaviors in off-policy RL.

We note that while we maintain full policy networks for each member of the population, it is likely possible to implement the population with shared initial layers but multiple heads (as in [39]). Space was not an issue in our case, but may be more important when using DvD for larger tasks (e.g. training from pixels). Furthermore, it is not a "free lunch", as the mean wall-clock time does tend to be higher for DvD (Fig. 7, second from right). However, this is often less important than sample efficiency in RL. Finally, the setting of $M$ is likely important. If too high, each agent would only explore for a brief period of time, or risk harming sample efficiency. If too small, we may not reap the benefit of using the population to explore. We believe investigating this is exciting future work.

## 7 Conclusion and Future Work

In this paper we introduced DvD, a novel method for promoting diversity in population-based methods for reinforcement learning. DvD addresses the issue of cycling by utilizing a joint population update via determinants corresponding to ensembles of policies' embeddings. Furthermore, DvD adapts the reward-diversity trade off in an online fashion, which facilitates flexible and effective diversity. We demonstrated across a variety of challenging tasks that DvD not only finds diverse, high quality solutions but also manages to maintain strong performances in one-good-solution settings.

We are particularly excited by the variety of research questions which naturally arise from this work. As we have mentioned, we believe it may be possible to learn the optimal population size, which is essentially the number of distinct high performing behaviors which exist for a given task. We are also excited by the prospect of learning embeddings or kernels, such that we can learn *which dimensions* of the action space are of particular interest. This could possibly be tackled with latent variable models.

## Acknowledgements and Disclosure of Funding

This research was made possible thanks to an award from the GCP research credits program. The authors want to thank anonymous reviewers for constructive feedback which helped improve the paper.

## Broader Impact

We believe our approach could be relevant for a multitude of settings, from population-based methods [32] to ensembles of models [39, 29]. There are two benefits to increased diversity:

1. The biggest short term advantage is performance gains for existing methods. We see improved sample efficiency and asymptotic gains from our approach to diversity, which may benefit any application with a population of agents [32] or ensemble of models. This may be used for a multitude of reasons, and one would hope the majority would be positive, such as model-based reinforcement learning for protein folding [2] (with an ensemble).

2. Having more diverse members of an ensemble may improve generalization, since it reduces the chance of models overfitting to the same features. This may improve robustness, helping real-world applications of reinforcement learning (RL). It may also lead to fairer algorithms, since a diverse ensemble may learn to make predictions based on a broader range of characteristics.

For reinforcement learning specifically, we believe our behavioral embeddings can be used as the new standard for novelty search methods. We have shown these representations are robust to design choices, and can work across a variety of tasks without domain knowledge. This counteracts a key weakness of existing novelty search methods, as noted by [26]. In addition, we are excited by future work building upon this, potentially by learning embeddings (as in [16]).

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
