[Supplementary Material]

# Appendix: Effective Diversity in Population Based Reinforcement Learning

## 8 Additional Experiment Details

### 8.1 Ablation Studies

Here we seek to analyze the sensitivity of DvD to design choices made, in order to gain confidence surrounding the robustness of our method.

**How sensitive is the embedding to the choice of states?** One of the crucial elements of this work is task-agnostic behavioral embedding. Similar to what has been used in all trust-region based policy gradient algorithms [52, 53], we use a concatenation of actions to represent policy behavior. In all experiments we used this behavioral embedding for both DvD and NSR, thus rendering the only difference between the two methods to be adaptive vs. fixed diversity and joint vs. individual updates.

However, there is still a question whether the design choices we made had an impact on performance. As such, we conducted ablation studies over the number $n$ of states chosen and different strategies for choosing states (for $n = 20$) used to construct behavioral embeddings.

|     |     |     |     |
|-----|-----|-----|-----|
| (a) **Point** | (b) **Swimmer** | (c) **Point** | (d) **Swimmer** |

Figure 8: The median best performing agent across five seeds. In a) and b) we vary the number of states selected in the random sample, in c) and d) we select 20 states but using different mechanisms. $\mathrm{Random}$ corresponds to uniform sampling, $\mathrm{Zero}$ to a zero function trained on all seen states, where we select the maximum variance states. DPP stands for a k-DPP [27].

As we see in Fig. 8, both the number of states chosen and the mechanism for choosing them appear to have minimal impact on the performance. In all cases the point escapes the local maximum (of moving into the area surrounded by walls) and Swimmer reaches high rewards ($> 300$).

**Choice of kernel** In Fig. 9 we include the plots accompanying Table 1. In almost all cases the performance is strong, and similar to the Squared Exponential kernel used in the main experiments.

|     |     |     |
|-----|-----|-----|
| (b) **Point** | (c) **Swimmer** | (d) **Walker2d** |

Figure 9: In this figure we show the median maximum performing agent across five seeds. The only difference between the two curves is the choice of kernel for the behavioral similarity matrix.

**What if the population size is much larger than the number of modes?** We also studied the relation between population size and the number of modes learned by the population. Both the tasks considered here have two modes, and intuitively a population of size $M = 3$ should be capable of learning both of them. However, we also considered the case for $M = 5$ (Fig. 10), and found that in fact a larger population was harmful for the performance. This leads to the interesting future work of adapting not only the degree of diversity but the size of the population.

(a) **Cheetah: Forward**      (b) **Cheetah: Backward**

Figure 10: Ablation study for the DvD algorithm: population size. The median best performing agent across ten seeds for the Cheetah multi-task environment. The only difference is the population sizes of $M = 3$ and $M = 5$.

## 8.2 Hyperparameters

**DvD-ES** In Table 2 we show the method for generating behavioral embeddings used for NSR-ES and our method. We used these settings across **all** experiments, and did not tune them. Number of states corresponds to the number of states used for the embedding, which is the concatenation of the actions from a policy on those states. State selection refers to the mechanism for selecting the states from the buffer of all states seen on the previous iteration. The update frequency is how frequently we re-sample states to use for the embeddings. We do not want this to be too frequent or we will have differnet embeddings every iteration simply as a result of the changing states.

Table 2: Configuration for the behavioral embedding, used across all experiments for our method and NSR-ES.

|  | Value |
|---|---|
| Number of states | 20 |
| State selection | random |
| Update frequency | 20 |

In Table 3 we show the hyprparameters used for the multi-modal experiments. We note the main difference between the two environments is the Ant requires more ES sensings (300 vs. 100) and a larger neural network (64-64 vs. 32-32) than the Cheetah. These settings were used for all three algorithms studied.

Table 3: Parameter configurations for the multi-modal experiments.

|  | **Cheetah** | **Ant** |
|---|---|---|
| $\sigma$ | 0.001 | 0.001 |
| $\eta$ | 0.001 | 0.001 |
| $h$ | 32 | 64 |
| ES-sensings | 100 | 300 |
| State Filter | True | True |

In Table 4 we show the hyperparameters used for the uni-model and deceptive reward experiments. The main difference is the size of the neural network for point and Swimmer is smaller (16-16 vs. 32-32) since these environments have smaller state and action dimensions than the others. In addition, we note the horizon $H$ for the point is smaller, as 50 timesteps is sufficient to reach the goal. These settings were used for all three algorithms considered.

Table 4: Parameter configurations for the single mode experiments. The only difference across all tasks was a smaller neural network used for Swimmer and point, since they have a smaller state and action dimensionality.

|  | **point** | **Swimmer** | **HalfCheetah** | **Walker2d** | **BipedalWalker** |
|---|---|---|---|---|---|
| $\sigma$ | 0.1 | 0.1 | 0.1 | 0.1 | 0.1 |
| $\eta$ | 0.05 | 0.05 | 0.05 | 0.05 | 0.05 |
| $h$ | 16 | 16 | 32 | 32 | 32 |
| ES-sensings | 100 | 100 | 100 | 100 | 100 |
| State Filter | True | True | True | True | True |
| $H$ | 50 | 1000 | 1000 | 1000 | 1600 |

**DvD-TD3** Our TD3 implementation comes from the following open source repository: .

All parameters are the same as in the original TD3 paper [15], apart from neural network architectures and choice of learning rate and batch size, which mirror SAC [19].

# 9 Theoretical Results

## 9.1 Proof of Theorem 3.3

*Proof.* We start by recalling that $\text{Div}(\Theta) = \det(\mathbf{K})$. Let $\alpha_1, \cdots, \alpha_M$ be the eigenvalues of $\mathbf{K}$. Since $\mathbf{K}$ is PSD, $\alpha_i \geq 0$ for all $i$. The following bounds hold:

$$0 \overset{(i)}{\leq} \text{Div}(\Theta) = \prod_{i=1}^{M} \alpha_i \overset{(ii)}{\leq} \left( \frac{\sum_{i=1}^{M} \alpha_i}{M} \right)^M = \left( \frac{\text{trace}(\mathbf{K})}{M} \right)^M \overset{(iii)}{=} 1$$

Inequality $(i)$ follows since $\mathbf{K}$ is a PSD matrix. Inequality $(ii)$ is a consequence of the AM-GM inequality and Equality $(iii)$ follows because all the diagonal entries of $\mathbf{K}$ equal 1. Let $\{\tilde{\pi}_i\}_{i=1}^{M}$ be a set of policies with at least one suboptimal policy and parametrized by $\tilde{\Theta}_t$. Wlog let $\mathcal{R}(\tilde{\pi}_1) + \Delta < \mathcal{R}$. The following holds:

$$\sum_{i=1}^{M} \mathcal{R}(\tilde{\pi}_i) + \lambda_t \text{Div}(\tilde{\Theta}_t) \leq M\mathcal{R} - \Delta + \lambda_t$$

Now observe that for any set of optimal policies $\{\pi_i\}_{i=1}^{M}$ parametrised by $\Theta_t$ the objective in Equation 7 satisfies:

$$\sum_{i=1}^{M} \mathcal{R}(\pi_i) + \lambda_t \text{Div}(\Theta_t) \geq M\mathcal{R}$$

Therefore if $\lambda_t < \Delta$, then:

$$\sum_{i=1}^{M} \mathcal{R}(\tilde{\pi}_i) + \lambda_t \text{Div}(\tilde{\Theta}_t) < \sum_{i=1}^{M} \mathcal{R}(\pi_i) + \lambda_t \text{Div}(\Theta_t)$$

Thus we conclude the objective in Equation 7 can only be maximised when all policies parametrised by $\Theta_t$ are optimal. Since $\lambda_t > 0$ and $\text{Div}(\Theta_t)$ is nonzero only when $\{\pi_i\}_{i=1}^{M}$ are distinct, and there exist at least $M$ distinct solutions, we conclude that whenever $0 < \lambda_t < \Delta$, the maximizer for Equation 7 corresponds to $M$ distinct optimal policies.

$\square$

## 9.2 Proof of Lemma 3.3

In the case of deterministic behavioral embeddings, it is possible to compute gradients through the whole objective. Notice that $\det(\mathbf{K})$ is then differentiable as a function of the policy parameters. In the case of trajectory based embeddings, differentiating through the determinant is not that simple. Actually it may make sense in this case to use a $\log(\det(\mathbf{K}))$ score instead. This is because of the following lemma:

**Lemma 9.1.** *The gradient of* $\log(\det(\mathbf{K}))$ *with respect to* $\theta = \theta^1, \cdots, \theta^M$ *equals:*

$$\nabla_\theta \log(\det(\mathbf{K})) = -(\nabla_\theta \Phi(\theta))(\nabla_\Phi \mathbf{K}) \mathbf{K}^{-1}$$

*Where* $\Phi(\theta) = \Phi(\theta^1) \cdots \Phi(\theta^M)$

*Proof.* We start with:

$$\nabla_{\mathbf{K}} \log(\det(\mathbf{K})) = -\mathbf{K}^{-1}$$

This is a known result[1].

Consequently,

$$\nabla_\theta \log\left(\det(\mathbf{K})\right) = \left(\nabla_\theta \Phi(\theta)\right)\left(\nabla_\Phi \mathbf{K}\right)\left(\nabla_{\mathbf{K}}\log\left(\det(\mathbf{K})\right)\right)$$
$$= -\left(\nabla_\theta \Phi(\theta)\right)\left(\nabla_\Phi \mathbf{K}\right)\mathbf{K}^{-1}$$

Each of the other gradients can be computed exactly.

$\square$

## 9.3 Determinants vs. Distances

In this section we consider the following question. Let $k(\mathbf{x}, \mathbf{y}) = \exp(-\|x-y\|^2)$. And $\mathbf{K} \in \mathbb{R}^{M \times M}$ be the kernel matrix corresponding to $M$ agents and resulting of computing the Kernel dot products between their corresponding embeddings $\{\mathbf{x}_1, \cdots, \mathbf{x}_M\}$:

**Theorem 9.2.** *For $M \leq 3$, the first order approximation of $\det(\mathbf{K})$ is proportional to the sum of the pairwise distances between $\{\mathbf{x}_1, \cdots, \mathbf{x}_M\}$. For $M > 3$ this first order approximation equals $0$.*

*Proof.* Consider the case of a population size $M = 3$, some policy embedding $\phi_i$ and the exponentiated quadratic kernel. In this setting, the diversity, measured by the determinant of the kernel (or *similarity*) matrix is as follows:

$$
\begin{aligned}
\det(\mathbf{K}) &= \begin{vmatrix} 1 & k(\mathbf{x}_1, \mathbf{x}_2) & k(\mathbf{x}_1, \mathbf{x}_3) \\ k(\mathbf{x}_2, \mathbf{x}_1) & 1 & k(\mathbf{x}_2, \mathbf{x}_3) \\ k(\mathbf{x}_3, \mathbf{x}_1) & k(\mathbf{x}_3, \mathbf{x}_2) & 1 \end{vmatrix} \\
&= 1 - k(\mathbf{x}_2, \mathbf{x}_3)k(\mathbf{x}_3, \mathbf{x}_2) - k(\mathbf{x}_1, \mathbf{x}_2)\big(k(\mathbf{x}_2, \mathbf{x}_1) - k(\mathbf{x}_3, \mathbf{x}_1)k(\mathbf{x}_2, \mathbf{x}_3)\big) + \\
&\quad k(\mathbf{x}_1, \mathbf{x}_3)\big(k(\mathbf{x}_2, \mathbf{x}_1)k(\mathbf{x}_3, \mathbf{x}_2) - k(\mathbf{x}_3, \mathbf{x}_1)\big) \\
&= 1 - k(\mathbf{x}_1, \mathbf{x}_2)^2 - k(\mathbf{x}_1, \mathbf{x}_3)^2 - k(\mathbf{x}_2, \mathbf{x}_3)^2 + 2k(\mathbf{x}_1, \mathbf{x}_2)k(\mathbf{x}_1, \mathbf{x}_3)k(\mathbf{x}_2, \mathbf{x}_3)
\end{aligned}
$$

So if we take $k$ to be the squared exponential kernel:

$$
\begin{aligned}
=&1 - \exp\left(\frac{-\|x_1 - x_2\|^2}{l}\right) - \exp\left(\frac{-\|x_1 - x_3\|^2}{l}\right) - \exp\left(\frac{-\|x_2 - x_3\|^2}{l}\right) + \\
&2\exp\left(\frac{-\|x_1 - x_2\|^2 - \|x_1 - x_3\|^2 - \|x_2 - x_3\|^2}{2l}\right)
\end{aligned}
$$

Recall that for $|x| << 1$ small enough, $\exp(x) \approx 1 + x$. Substituting this approximation in the expression above we see:

$$
\begin{aligned}
\det(\mathbf{K}) &\approx \|x_1 - x_2\|^2 + \|x_1 - x_3\|^2 + \|x_2 - x_3\|^2 - \frac{\|x_1 - x_2\|^2 + \|x_1 - x_3\|^2 + \|x_2 - x_3\|^2}{2} \\
&= \frac{\|x_1 - x_2\|^2 + \|x_1 - x_3\|^2 + \|x_2 - x_3\|^2}{2},
\end{aligned}
$$

which is essentially the mean pairwise $l_2$ distance. What can we say about these differences (e.g. $\exp$ vs. not)? Does this same difference generalize to $M > 3$?

**Approximation for $M > 3$**

Recall that for a matrix $\mathbf{A} \in \mathbb{R}^{M \times M}$, the determinant can be written as:

$$\det(\mathbf{A}) = \sum_{\sigma \in \mathbb{S}_M} \text{sign}(\sigma) \prod_{i=1}^{M} \mathbf{A}_{i, \sigma(i)}$$

Where $\mathbb{S}_M$ denotes the symmetric group over $M$ elements. Lets identify $A_{i,j} = \exp\left(-\frac{\|x_i-x_j\|^2}{2}\right)$. Notice that for any $\sigma \in \mathbb{S}_M$, we have the following approximation:

$$\prod_{i=1}^{M} \mathbf{A}_{i,\sigma(i)} \approx 1 - \sum_{i=1}^{M} \frac{\|x_i - x_{\sigma(i)}\|^2}{2} \tag{11}$$

Whenever for all $i, j \in [M]$ the value of $\|x_i - x_j\|^2$ is small.

We are interested in using this termwise approximation to compute an estimate of $\det(\mathbf{A})$. Plugging the approximation in Equation 11 into the formula for the determinant yields the following:

$$\det(\mathbf{A}) \approx \sum_{\sigma \in \mathbb{S}_M} \text{sign}(\sigma)\left(1 - \sum_{i=1}^{M} \frac{\|x_i - x_{\sigma(i)}\|^2}{2}\right)$$

$$= \underbrace{\sum_{\sigma \in \mathbb{S}_M} \text{sign}(\sigma)}_{\text{I}} - \underbrace{\sum_{\sigma \in \mathbb{S}_M} \text{sign}(\sigma)\sum_{i=1}^{M} \frac{\|x_i - x_{\sigma(i)}\|^2}{2}}_{\text{II}}$$

Term I equals zero as it is the sum of all signs of the permutations of $\mathbb{S}_n$ and $n > 1$.

In order to compute the value of II we observe that by symmetry:

$$\text{II} = B\sum_{i<j} \|x_i - x_j\|^2$$

For some $B \in \mathbb{R}$. We show that $B = 0$ for $M > 3$. Let's consider the set $B_{1,2}$ of permutations $\sigma \in \mathbb{S}_M$ for which the sum $\sum_{i=1}^{M} \frac{\|x_i - x_{\sigma(i)}\|^2}{2}$ contains the term $\frac{\|x_1-x_2\|^2}{2}$. Notice that $B = \frac{1}{2}\sum_{\sigma \in B_{1,2}} \text{sign}(\sigma)$. Let's characterize $B_{1,2}$ more exactly.

Recall every permutation $\sigma \in \mathbb{S}_M$ can be thought of as a product of cycles. For more background on the cycle decomposition of permutations see [57].

The term $\frac{\|x_1-x_2\|^2}{2}$ appears whenever the cycle decomposition of $\sigma$ contains a transition of the form $1 \to 2$ or $1 \leftarrow 2$. It appears twice if the cycle decomposition of $\sigma$ has the cycle corresponding to a single transposition $1 \leftrightarrow 2$.

Let $\overrightarrow{B}_{1,2}$ be the set of permutations containing a transition of the form $1 \to 2$ (and no transition of the form $1 \leftarrow 2$) $\overleftarrow{B}_{1,2}$ be the set of permutations containing a transition of the form $1 \leftarrow 2$ (and no transition of the form $1 \to 2$) and finally $\overleftrightarrow{B}_{1,2}$ be the set of permutations containing the transition $1 \leftrightarrow 2$.

Notice that:

$$B = \underbrace{\sum_{\sigma \in \overrightarrow{B}_{1,2}} \text{sign}(\sigma)}_{O_1} + \underbrace{\sum_{\sigma \in \overleftarrow{B}_{1,2}} \text{sign}(\sigma)}_{O_2} + 2\underbrace{\sum_{\sigma \in \overleftrightarrow{B}_{1,2}} \text{sign}(\sigma)}_{O_3}$$

We start by showing that for $M > 3$, $O_3 = 0$. Indeed, any $\sigma \in \overleftrightarrow{B}_{1,2}$ has the form $\sigma = (1,2)\sigma'$ where $\sigma'$ is the cycle decomposition of a permutation over $3, \cdots, M$. Consequently $\text{sign}(\sigma) = -\text{sign}(\sigma')$. Iterating over all possible $\sigma' \in \mathbb{S}_{M-2}$ permutations over $[3, \cdots, M]$ yields the set $\overleftrightarrow{B}_{1,2}$ and therefore:

$$O_3 = -\sum_{\sigma' \in \mathbb{S}_{M-2}} \text{sign}(\sigma')$$

$$= 0$$

The last equality holds because $M - 2 \geq 2$. We proceed to analyze the terms $O_1$ and $O_2$. By symmetry it is enough to focus on $O_1$. Let $c$ be a fixed cycle structure containing the transition $1 \to 2$. Any $\sigma \in \overrightarrow{B}_{1,2}$ containing $c$ can be written as $\sigma = c\sigma'$ where $\sigma'$ is a permutation over the remaining elements of $\{1, \cdots, M\} \backslash c$ and therefore $\text{sign}(\sigma) = (-1)^{|c|-1}\text{sign}(\sigma')$. Let $\overrightarrow{B}_{1,2}^c$ be the subset of $\overrightarrow{B}_{1,2}$ containing $c$.

Notice that:

$$O_1 = \sum_{c|1\to2\in c,|c|\geq 3} \underbrace{\left[ \sum_{\sigma\in\overrightarrow{B}_{1,2}^c} \text{sign}(\sigma) \right]}_{O_1^c}$$

Let's analyze $O_1^c$:

$$O_1^c = \sum_{\sigma\in\overrightarrow{B}_{1,2}^c} \text{sign}(\sigma)$$
$$= (-1)^{|c|-1} \sum_{\sigma\in\mathbb{S}_{M-|c|}} \text{sign}(\sigma)$$

If $|\{1, \cdots, M\} \backslash c| \geq 2$ this quantity equals zero. Otherwise it equals $(-1)^{|c|-1}$. We recognize two cases, when $|\{1, \cdots, M\} \backslash c| = 0$ and when $|\{1, \cdots, M\} \backslash c| = 1$ . The following decomposition holds [2] :

$$O_1 = \left|\{c|1 \to 2 \in c, |c| \geq 3|\{1, \cdots, M\}\backslash c| = 0\}\right| * (-1)^{M-1} +$$
$$|\{c|1 \to 2 \in c, |c| \geq 3|\{1, \cdots, M\}\backslash c| = 1\}| * (-1)^{M-2}$$

A simple combinatorial argument shows that:

$$|\{c|1 \to 2 \in c, |\{1, \cdots, M\}\backslash c| = 0\}| = (M - 2)!$$

Roughly speaking this is because in order to build a size $M$ cycle containing the transition $1 \to 2$, we only need to decide on the positions of the next $M - 2$ elements, which can be shuffled in $M - 2$ ways. Similarly, a simple combinatorial argument shows that:

$$|\{c|1 \to 2 \in c, |\{1, \cdots, M\}\backslash c| = 1\}| = (M - 2)!$$

A similar counting argument yields this result. First, there are $M - 2$ ways of choosing the element that will not be in the cycle. Second, there are $(M - 3)!$ ways of arranging the remaining elements to fill up the $M - 3$ missing slots of the $M - 1$ sized cycle $c$.

We conclude that in this case $O_1 = 0$.

$\square$

This result implies two things:

1. When $M \leq 3$. If the gradient of the embedding vectors is sufficiently small, the determinant penalty is up to first order terms equivalent to a pairwise distances score. This may not be true if the embedding vector's norm is large.

2. When $M > 3$. The determinant diversity penalty variability is given by its higher order terms. It is therefore not equivalent to a pairwise distances score.

# 10 Extended Background

For completeness, we provide additional context for existing methods used in the paper.

## 10.1 Thompson Sampling

Let's start by defining the Thompson Sampling updates for Bernoulli random variables. We borrow the notation from Section 4.2. Let $\mathcal{K} = \{1, \cdots, K\}$ be a set of Bernoulli arms with mean parameters $\{\mu_i\}_{i=1}^K$.

Denote by $\pi_t^i$ the learner's mean reward model for arm $i$ at time $t$. We let the learner begin with an independent prior belief over each $\mu_i$, which we denote $\pi_o^i$. These priors are beta-distributed with parameters $\alpha_i^o = 1, \beta_i^o = 1$:

$$\pi_o^i(\mu) = \frac{\Gamma(\alpha_i^o + \beta_i)}{\Gamma(\alpha_i^o)\Gamma(\beta_i)}\mu^{\alpha_i^o - 1}(1 - \mu)^{\beta_i^o - 1},$$

Where $\Gamma$ denotes the gamma function. It is convenient to use beta distributions because of their conjugacy properties. It can be shown that whenever we use a beta prior, the posterior distribution is also a beta distribution. Denote $\alpha_i^t, \beta_i^t$ as the values of parameters $\alpha_i, \beta_i$ at time $t$.

Let $i_t$ be the arm selected by Thomson Sampling, as we explained in main body, at time $t$. After observing reward $r_t \in \{0, 1\}$ the arms posteriors are updated as follows:

$$(\alpha_i^{t+1}, \beta_i^{t+1}) = \begin{cases} (\alpha_i^t + r_t, \beta_i^t + (1 - r_t)) & \text{if } i = i_t \\ (\alpha_i^t, \beta_i^t) & \text{o.w.} \end{cases}$$

## 10.2 Reinforcement Learning Algorithms

### 10.2.1 Evolution Strategies

ES methods cast RL as a blackbox optimization problem. Since a blackbox function $F : \mathbb{R}^d \to \mathbb{R}$ may not even be differentiable, in practice its smoothed variants are considered. One of the most popular ones, the Gaussian smoothing [38] $F_\sigma$ of a function $F$ is defined as:

$$F_\sigma(\theta) = \mathbb{E}_{\mathbf{g} \in \mathcal{N}(0, \mathbf{I}_d)}[F(\theta + \sigma\mathbf{g})]$$
$$= (2\pi)^{-\frac{d}{2}} \int_{\mathbb{R}^d} F(\theta + \sigma\mathbf{g})e^{-\frac{\|\mathbf{g}\|^2}{2}} d\mathbf{g},$$

where $\sigma > 0$ is a hyperparameter quantifying the smoothing level. Even if $F$ is nondifferentiable, we can easily obtain stochastic gradients for $F_\sigma$. The gradient of the Gaussian smoothing of $F$ is given by the formula:

$$\nabla F_\sigma(\theta) = \frac{1}{\sigma}\mathbb{E}_{\mathbf{g} \sim \mathcal{N}(0, \mathbf{I}_d)}[F(\theta + \sigma\mathbf{g})\mathbf{g}]. \tag{12}$$

This equation leads to several Monte Carlo gradient estimators used successfully in Evolution Strategies (ES, [49, 7]) algorithms for blackbox optimization in RL. Consequently, it provides gradient-based policy update rules such as:

$$\theta_{t+1} = \theta_t + \eta\frac{1}{k\sigma}\sum_{i=1}^k R_i\mathbf{g}_i, \tag{13}$$

where $R_i = F(\theta_t + \sigma\mathbf{g}_i)$ is the reward for perturbed policy $\theta_t + \sigma\mathbf{g}_i$ and $\eta > 0$ stands for the step size.

In practice Gaussian independent perturbations can be replaced by dependent ensembles to further reduce variance of the Monte Carlo estimator of $\nabla F_\sigma(\theta)$ via quasi Monte Carlo techniques.

### 10.2.2 Novelty Search

In the context of population-based Reinforcement Learning, one prominent approach is the class of novelty search methods for RL [30, 31, 9]. The NSR-ES algorithm [9] maintains a meta-population

of $M$ agents, and at each iteration $t$ sequentially samples and an individual member $\theta_t^m$. This agent is perturbed with samples $\mathbf{g}_1^m, \cdots, \mathbf{g}_k^m \sim \mathcal{N}(0, \mathbf{I}_d)$, and then the rewards $R_i^m = F(\theta_t^m + \sigma \mathbf{g}_i^m)$ and embeddings $\Phi(\theta_t^m + \sigma \mathbf{g}_i^m)$ are computed in parallel. The *novelty* of a perturbed policy is then computed as the mean Euclidean distance of its embedding to the embeddings $\Phi(\theta_t^i)$ of the remaining members of the population for $i \neq m$. In order to update the policy, the rewards and novelty scores are normalized (denoted $\widehat{R}_i^m$ and $\widehat{N}_i^m$), and the policy is updated as follows:

$$\theta_{t+1}^m = \theta_t^m + \frac{\eta}{k\sigma} \sum_{i=1}^{k} [(1-\lambda)\widehat{R}_i^m + \lambda\widehat{N}_i^m]\mathbf{g}_i, \tag{14}$$

where the novelty weight $\lambda > 0$ is a hyperparameter. A value $\lambda = 0$ corresponds to the standard ES approach (see: Eq. 13) whereas the algorithm with $\lambda = 1$ neglects the reward-signal and optimizes solely for diversity. A simple template of this approach, with a fixed population size, appears in Alg. 1.

Despite encouraging results on hard exploration environments, these algorithms contain several flaws. They lack a rigorous means to evaluate the diversity of the population as a whole, this means that $M$ policies may fall into $N < M$ conjugacy classes, leading to the illusion of a diverse population on a mean pairwise Euclidean distance metric.

---

**Algorithm 1** Population-Based NSR-ES

---

**Input:** : learning rate $\eta$, noise standard deviation $\sigma$, number of policies to maintain $M$, number of iterations $T$, embedding $\Phi$, novelty weight $\lambda$.
**Initialize:** $\{\theta_0^1, \ldots, \theta_0^M\}$.
**for** $t = 0, 1, \ldots, T-1$ **do**
    1. Sample policy to update: $\theta_t^m \sim \{\theta_t^1, \ldots, \theta_t^M\}$.
    2. Compute rewards $F(\theta_t^m + \sigma\mathbf{g}_k)$ for all $\mathbf{g}_1, \cdots, \mathbf{g}_k$, sampled independently from $\mathcal{N}(0, \mathbf{I}_d)$.
    3. Compute embeddings $\Phi(\theta_t^m + \sigma\mathbf{g}_k)$ for all $k$.
    4. Let $\widehat{R}_k$ and $\widehat{N}_k$ be the normalized reward and novelty for each perturbation $\mathbf{g}_k$.
    5. Update Agent via Equation 14.

---

## Footnotes

[1]see for example `https://math.stackexchange.com/questions/38701/how-to-calculate-the-gradient-of-log-det-matrix-inverse`

[2] Here we use the assumption $M \geq 4$ to ensure that both $|\{c|1 \to 2 \in c, |c| \geq 3|\{1, \cdots, M\}\backslash c| = 0\}| > 0$ and $|\{c|1 \to 2 \in c, |c| \geq 3|\{1, \cdots, M\}\backslash c| = 1\}| > 0$.