[Reviews · NeurIPS 2020]

Review 1

Summary and Contributions: The paper offers a method to improve exploration by training a group of agents with diverse behaviors. The authors seek to train a group of agents to cover a maximum area of the behavioral manifold that defines the set of possible policies. The authors present two variants of their method. One that is based on evolution strategies and one that is based on gradient descent. Besides that, the authors use Thompson sampling to control the degree of regularization by simultaneously solving a multi-armed bandit problem over a discrete set of regularization coefficients. The authors evaluate their method in a set of continuous conrol challenges from the Mujoco benchmark and show improvement over NSR- an evolution strategy algorithm.

Strengths: The paper identifies and addresses an important problem that arises when handling ensembles of agents. The naive requirement of pairwise diversity indeed do not lead to global diversity. The authors include a toy MDP example in section 3.3 that well exemplifies their claim.

Weaknesses: 1. Not only that the motivation of Section 3.1 seems irrelevant to the method, but it also creates unnecessary confusion (the motivation talks about cross-entropy RL, i.e., trust-region methods). While (minimum) cross-entropy RL is a stabilization framework, this paper is about diversification and exploration which is quite the opposite. I think that the motivation confuses between minimum cross entropy RL (stability), and maximum cross entropy RL (exploration/diversity). 2. In my opinion, the most interesting part of this work is to see how the authors derive an optimization algorithm from the determinant regularization (in the gradient-based version). However, this was deferred to the appendix (lemma 4.1), but the appendix is missing. 3. The result on Figure 5 is weird. The authors claim that they do not hurt performance when exploration is not needed. However, this can be contributed to the Thompson smapling part that probably turned exploration off (lambda=0). Ablation studies are missing to understand what is going on. 4. The proposed method comprises of two major modules: determinant regularization, and Thompson sampling of the regularization’s coefficient. Since the two modules are “orthogonal” it is necessary to provide ablation tests in this case. While reviling that the former module is responsible for most of the improvement is interesting, seeing that the improvement comes from the Thompson sampling part makes the results in this paper somewhat less significant.

Correctness: I am not able to evaluate the correctness of the results because all proofs are deferred to a missing appendix. Moreover, the gradient of the regularization term includes a matrix inversion and I am curious to understand how this is carried out in practice.

Clarity: Unfortunately not. The paper should undergo serious proofreading. Listed here is a partial list of typos and spelling errors. 1. line 23: “Training a population of agents provides is a promising approach” 2. line 45: “we note that it is remains” 3. line 68: “targeting learning a value function” 4. line 78: “as taking as input parameters” 5. line 232: “it is proposed to use of a population of agents”

Relation to Prior Work: The authors mention the MAP-Elite aslgorithm as a similar method to their work. However the algorithm is not present in the evaluation part. It would be interesting to see a comparison with this method.

Reproducibility: No

Additional Feedback: Preliminaries section: the section about different RL approaches is too elaborated and can be shortened to free space for important results deferred to the appendix.


Review 2

Summary and Contributions: The main contribution of this paper is to propose a novel method for promoting diversity for direct policy search in RL using gradient-free methods, what the authors call population-based methods. Their method consists in adding to the loss function a population diversity term based on a kernel defining a distance between two policies. They also have a proof showing that their approach indeed promotes diversity, even if the result is not that strong.

Strengths: The paper is well-written and pretty clear. They have theoretical results. The simulations appear to be carried out in a competent way and highlight the interest of their approach.

Weaknesses: The approach they propose is itself not that original. Minor remark that I would like to be addressed: It bothers me that you are considering stationary policies in a finite time setting. They are suboptimal

Correctness: Yes, I would say so.

Clarity: Yes.

Relation to Prior Work: Yes

Reproducibility: Yes

Additional Feedback:


Review 3

Summary and Contributions: The paper proposes a way of measuring diversity of behavior for a population of agents. This is useful because many RL techniques run several agents in parallel to more efficiently learn optimal policies, but in many tasks they can 'stuck' in local optima. This measure of diversity of policies is incorporated into the optimization objective, encouraging agents' behavior to be diverse; thus, possibly escaping local optima.

Strengths: This is a very relevant problem to the RL community, specially for tasks that can be simulated. Diversity of behaviors can help agents escape local optima, or discover new skills, which in turn leads to finding solution that could not have been found otherwise. The paper is well-written, formalizes the problem clearly, and the results are encouraging.

Weaknesses: The main weakness of this paper, in my opinion, is the experimental section. Many details for each experiment are missing, and they feel a bit 'rush'. My suggestion would be to select fewer experiments and have a more felshed out analysis of them. For example: in the Exploration task, the authors state NSR-ES fails to solve this problem, but DvD is able to solve it in all instances. Why is that the case? Since I am familiar with the topic, I can infer that the other methods got stuck in a local optimum hitting the wall, and adding diversity allowed some agents to reach policies that move around that wall and successfully reach the goal. The reader should not be left to infer these results. Another example is the Humanoid experiments. At first sight, the results do not look particularly meaningful; but the authors significance at a p <0.05 level based on Welch's t-test. The reader should have more context on how this t-test was performed. What are the distributions that are being compared, and why can we assume they are normal distributions? - What is the computational cost of taking the gradient of the new diversity term? The humanoid section mentions that the new method is able to reach a score of 6,000 in 1M steps while SAC needs about 3M. Is the update of the proposed approach hiding some costly computation, or is the cost of each update comparable between the two method. Whichever is true, should be mentioned to not leave the reader guessing.

Correctness: For the most part, although some details in empirical experiments are missing, or should be clarified. - Please respond to the t-test comment above. - How many trials where run per experiment? Exploration mentions 10 seeds, is that the case for all other experiments as well?

Clarity: Yes

Relation to Prior Work: As far as I can tell, it is well placed in the context of relevant literature.

Reproducibility: Yes

Additional Feedback: Besides the comments above, I have a few points where I would appreciate a bit of clarification. - line 73 mentions that the transition dynamics are often deterministic. This is actually rarely the case in most interesting problems. Did you mean to say stationary? - line 106, definition of behavioral embedding. Is the behavioral embedding a matrix where each row i corresponds to the prob distribution over actions for state i? Am I reading this correctly? - line 109, why are policies deterministic? Does this method not apply to stochastic policies? If so, it would be nice to state it right at the intro/abstract, so that the reader can know whether this applies or not to their problems. - ES and TD3, please introduce abbreviations the first time they are used. - Just personal preference, feel free to ignore if you don't agree. It threw me off a bit having a related work section before the experiments. Normally I would expect it after the introduction or right before the conclusion. ======= After rebuttal Thank you for taking the time to respond to our feedback. I like the idea in general, but in my opinion the experimental section could be improved quite a bit. As presented, it is difficult to judge how significant the results are.


Review 4

Summary and Contributions: This paper proposed a new algorithm called DvD that can promote behavioral diversities among all policies in a population. This is achieved through joint use of several key techniques, including behavioral embedding in the action space, Kernel-based behavioral comparison, population-wide diversity measure through calculating the determinant of the kernel matrix and Thompson sampling for adaptive updating of lambda.

Strengths: This paper introduces several interesting ideas. I found the idea of measuring population diversity through kernel function and determinant of the kernel matrix particularly interesting and novel. The research questions explored in the paper are also highly relevant to the NeurIPS community.

Weaknesses: The paper may need to be improved to address a few important issues, as detailed below. 1. Why is it important to enhance population-wide behavioral diversity? Intuitively I can understand the potential benefits related to deep exploration and learning stability. However, theoretically I cannot link the benefits straightforwardly to the proposed use of kernel function and the kernel matrix determinant. Theorem 3.3 states that when lambda is set properly, the population will contain M distinct optimal policies. This theorem assumes the existence of a positive constant Delta. This assumption is not always valid, especially upon considering stochastic policies. Even when the theorem is valid, why is it necessary to find all the distinct optimal policies when the learning goal is just to find one of them? In other words, why, by trying to find all optimal policies, the samples collected by DvD will increase the chance for the learning algorithm to escape from local optima and achieve higher sample efficiency, as demonstrated in Figure 6? This paper provided some illustrating examples on page 4 and page 5. However, I am still not fully convinced by the necessity of finding all optimal policies.  As further noted by the authors in the appendix, if the population size is larger than the number of modes, the learning performance will be negatively affected. In view of this, how many optimal policies that DvD should search for simultaneously in a population? Why collecting samples by using more policies in the population will actually affect the learning performance, provided that we can maintain good diversity among all policies? 2. After checking the proof of Theorem 3.3 in the appendix, I cannot understand the last inequality on page 15. Given the condition that R(pi)+Delta<R and Div(Theta)<1, the LHS of the inequality should be less than M*R-M*Delta+lambda rather than M*R-Delta+lambda. The correctness of this step of derivation may need to be double checked. Also for the kind of learning problems studied in the paper, the condition on the existence of positive constant Delta may not be satisfied. Hence the theorem may not be able to explain how the diversity measure can actually drive the learning of all optimal policies. 3. The proposed behavioral embedding requires a finite set of states. However, for many practical problems, the state space is not only infinite but also uncountable. In this case, the authors proposed to use sampled states to estimate the behavioral embedding. I was wondering how such sampled embedding may affect the diversity of the population and the performance of DvD. Some theoretical analysis would be very helpful for me to truly understand the importance and novelty of the newly proposed behavioral embedding. Similarly, some high-level intuitive discussions were presented on page 4 to justify the use of kernel matrix determinant as a measure of population diversity. In addition to high-level discussions, I think detailed analysis is also required to clearly justify the efficacy of using the determinant measure, especially when it may noticeably increase the computation complexity of the resulting algorithm. 4. The use of notations should be carefully managed. For example M is used in Theorem 3.3 to refer to an MDP. Meanwhile M is also used to represent the number of agents in the population. This may cause some confusion to the readers. 5. DvD introduces several new hyper-parameters, for example hyper-parameters related to the behavioral embeddings, the choice of the kernel function, the number of agents, as well as those highlighted in Table 3 and Table 4. How to determine the suitable values of these hyper-parameters? Is it easy to tune these hyper-parameters in practice? Are there any empirical rules to follow? More experiment results may need to be presented in order to understand the performance impact of these hyper-parameters. 6. Figure 6 does not seem to show that DvD can clearly outperform TD3. Although statistical test results indicate that the performance difference is significant, the test was performed on experiments that used only 7 seeds. Hence the test results may not be very reliable. Even DvD achieved higher performance than TD3, it is more complicated in design than TD3. How does DvD compare to TD3 in terms of computation cost? I think we need more evidences and more experiment results to truly understand the advantages of using DvD over TD3 and other cutting-edge deep reinforcement learning algorithms. Many thanks for the authors' feedback. I still feel several aspects of this paper may need to be improved with more details. Meanwhile I did not seem to be able to find any direct feedback with regard to the proof of Theorem 3.3 and the validity of its assumptions.

Correctness: Most of the claims and methods appear to be correct. The derivation of theorem 3.3 may need to be double checked. The validity of this theorem should be verified further and its connection to the improved learning performance should be justified more. The applicability of statistical tests on limited data may also need to be carefully checked.

Clarity: This paper is generally well written and easy to follow.

Relation to Prior Work: The connection to previous works has been presented in section 5. The key difference from previous research has also been elucidated.

Reproducibility: Yes

Additional Feedback:

[Author Response · NeurIPS 2020]

We would firstly like to thank the reviewers for their time. We understand many of the issues highlighted were largely caused by a lack of clarity. We humbly request for more of your time in re-evaluating scores, given our responses below.

**R1**: We are pleased to see you like the idea, and motivation. However, it seems the reject decision hinges on many results from a "missing" Appendix **which was indeed included**. In the supplementary materials, (DvD_Supplement.pdf) we included the proof for Lemma 4.1 (l. 584-596), as well as ablation studies for the different components of our method (l. 528-532). Given that the majority of your concerns were fully addressed in the Appendix, we would greatly appreciate it if you could please update your score to reflect this. In particular, Fig 8 shows that, while it is true the adaptive method outperforms the fixed positive diversity, in the latter case we can **still learn good policies**. This is not the case for NSR-ES (due to the cycling issues). The method with $\lambda = 0$ is already included in the main body, it is the vanilla ES. The purpose of Section 3.1 is to introduce the notion of behavioral embeddings which we use to define a similarity measure between policies and is crucial to the definition of our joint objective in equation (8). We agree with your comment (#1) that we should remove the confusing reference to Trust Region Methods in Section 3.1, this will save space for additional clarity elsewhere. Regarding a MAP-Elites comparison, typically when people use this algorithm they have a specific problem in mind so use domain knowledge to form the grid. Thus, we cannot compare against that approach. The matrix inversion is trivial as $\mathbf{K}$ is a $M \times M$ matrix ($M \in \{3, 5\}$).

**R2**: Thank you for your review, and comment that our paper is novel and well written. Regarding the use of stationary policies in a finite time setting, this is a standard approach in deep RL and beyond the scope of our paper.

**R3**: We appreciate your positive comments, from reading them, it was a surprise to see such a negative score (reject). In particular, you say "This is a very relevant problem to the RL community... The paper is well-written, formalizes the problem clearly, and the results are encouraging." When looking at the negative comments, it appears the issues are all things we can clarify with an extra page in the camera ready. The first comment claims we do not explain the behaviors for NSR-ES and ES. However, in the paper **we explicitly said this**, see l. 268 "As we see, both vanilla ES and NSR-ES fail to get past the wall (a reward of -800)". Second, regarding the t-test, we used the distribution of maximum rewards obtained per seed. We used a similar approach to multiple other studies. Of course, we can add additional comments that no t-test is perfect, however, it seems to be a minor remark and not one specific to our work. We are surprised it's used as one of three comments to reject our paper. Finally, the computational cost of the gradients. We do agree we should have included this. We ran these experiments on a laptop computer, without a GPU, and it used on average $27\%$ more wall-clock time to train with DvD. We believe this would be reduced on a GPU. We also note that in RL, we often care more about samples, which may be from the real world, and our results are a **new state of the art** in this context. We will absolutely include this discussion in the paper. The number of seeds is included in the caption for each figure. In this paper we focus on deterministic policies, we can include this earlier in the paper. It could be extended to stochastic policies if we used a kernel which measures similarity between distributions.

**R4**: Thank you for your detailed comments, we very much appreciate them and believe they open up exciting directions for future work. In particular, how do we know how many optimal policies to look for? This is currently a hyperparameter and ideally we would be able to learn the number of optimal solutions for a given problem. We could do this with an adaptive population size, for example, if there are only three possible optimal solutions we do not need a population of ten agents. However, this issue is common across all quality diversity algorithms and we believe it is beyond the scope (and length) of our paper. Next, *why* we wish to find diverse solutions (#1), there have been many different use cases for quality diversity algorithms. For example if the distribution shifts (the classic example from Cully 2015), we may wish to switch "behaviors", which means having access to a set of distinct (yet high performing) policies. In the examples we used, it is simply the case that by adding diversity you are able to explore better (motivated by Conti 2018). Outside of RL, there has been focus on creating diversity in deep ensembles, to improve uncertainty callibration on out of distribution data, an approach such as DvD may also make sense here for future work.

(#2) The inequality in the paper is correct - the argument assumes there is at least one suboptimal policy, not all of them. A well-known theorem in RL states that there is always a deterministic optimal policy. This justifies restricting ourselves to deterministic policies, albeit, with stochastic policies as future work. The reviewer is right, the assumption on $\Delta$ may not be always satisfied. Nevertheless it is a reasonable assumption for many classes of problems including sparse reward scenarios. (#4) Justifying the determinant: we feel this has been covered with the example. The determinant ensures no two solutions are the same, rather than just the **average** difference being high. Our theorem directly addresses this. Thank you for highlighting the notation issue, we will fix this. (#5) Regarding hyperparameters, we address many of these in the Appendix. We show the choice of kernel, and means to sample states for the embedding, do not have a huge impact. The key hyperparameter is the number of diverse solutions, which we previously discussed. (#6) See above for discussion of computational cost, we will include this in the paper. We feel this is not so much about DvD vs. TD3, but more that DvD can be used alongside TD3 to boost exploration.

[Meta-Review · NeurIPS 2020]

This paper focuses on an interesting problem of maintaining diversity in a set of agents. The paper formalizes the problem clearly, and the initial results presented are positive and support the paper's claims. The paper is fairly well written. Among the aspects of the paper that could be improved upon, the experimental section lacks many details and no ablation studies are performed, it is not clear how the new technique compares from a computational cost perspective, and some of the implications of the assumptions made (e.g., on the scope of problems to which this approach can reasonably be applied) are not clearly stated. Overall, the paper was found to make a sufficient contribution and the final recommendation is to accept. Please take reviewer comments into account as you are preparing the final version.